# The Marine Seagrass *Halophila stipulacea* as a Source of Bioactive Metabolites against Obesity and Biofouling

**DOI:** 10.3390/md18020088

**Published:** 2020-01-29

**Authors:** Sawssen Bel Mabrouk, Mariana Reis, Maria Lígia Sousa, Tiago Ribeiro, Joana R. Almeida, Sandra Pereira, Jorge Antunes, Filipa Rosa, Vitor Vasconcelos, Lotfi Achour, Adnen Kacem, Ralph Urbatzka

**Affiliations:** 1Research Laboratory (LR14-ES06): Bioresources: Integrative Biology and Valorization, High Institute of Biotechnology of Monastir, University of Monastir, Monastir 5000, Tunisia; iris1989@live.fr (S.B.M.); lotfiachour@yahoo.fr (L.A.); adnenkacem@yahoo.fr (A.K.); 2Interdisciplinary Centre of Marine and Environmental Research (CIIMAR/CIMAR), University of Porto, Avenida General Norton de Matos, s/n, 4450-208 Matosinhos, Portugal; mariana.a.reis@gmail.com (M.R.); ligiasilvasousa@gmail.com (M.L.S.); tribeiro@ciimar.up.pt (T.R.); joana.reis.almeida@gmail.com (J.R.A.); jorgetantunes@gmail.com (J.A.); frosa@ciimar.up.pt (F.R.); vmvascon@fc.up.pt (V.V.); 3Department of Biology, FCUP, Faculty of Science, University of Porto, Rua do Campo, Alegre, 4169-007 Porto, Portugal

**Keywords:** *Halophila stipulacea*, extracts, bioactivity, cytotoxicity, anti-obesity, antifouling

## Abstract

Marine organisms, including seagrasses, are important sources of biologically active molecules for the treatment of human diseases. In this study, organic extracts of the marine seagrass *Halophila stipulacea* obtained by different polarities from leaves (L) and stems (S) (hexane [HL, HS], ethyl acetate [EL, ES], and methanol [ML, MS]) were tested for different bioactivities. The screening comprehended the cytotoxicity activity against cancer cell lines grown as a monolayer culture or as multicellular spheroids (cancer), glucose uptake in cells (diabetes), reduction of lipid content in fatty acid-overloaded liver cells (steatosis), and lipid-reducing activity in zebrafish larvae (obesity), as well as the antifouling activity against marine bacteria (microfouling) and mussel larval settlement (macrofouling). HL, EL, HS, and ES extracts showed statistically significant cytotoxicity against cancer cell lines. The extracts did not have any significant effect on glucose uptake and on the reduction of lipids in liver cells. The EL and ML extracts reduced neutral lipid contents on the larvae of zebrafish with EC_50_ values of 2.2 µg/mL for EL and 1.2 µg/mL for ML. For the antifouling activity, the HS and ML extracts showed a significant inhibitory effect (*p* < 0.05) against the settlement of *Mytilus galloprovincialis* plantigrade larvae. The metabolite profiling using HR-LC-MS/MS and GNPS (The Global Natural Product Social Molecular Networking) analyses identified a variety of known primary and secondary metabolites in the extracts, along with some unreported molecules. Various compounds were detected with known activities on cancer (polyphenols: Luteolin, apeginin, matairesinol), on metabolic diseases (polyphenols: cirsimarin, spiraeoside, 2,4-dihydroxyheptadec-16-ynyl acetate; amino acids: N-acetyl-L-tyrosine), or on antifouling (fatty acids: 13-decosenamide; cinnamic acids: 3-hydroxy-4-methoxycinnamic acid, alpha-cyano-4-hydroxycinnamic), which could be, in part, responsible for the observed bioactivities. In summary, this study revealed that *Halophila stipulacea* is a rich source of metabolites with promising activities against obesity and biofouling and suggests that this seagrass could be useful for drug discovery in the future.

## 1. Introduction

The marine environment is an exceptional reservoir of new bioactive compounds, which often exhibit different structural and chemical features compared to terrestrial natural products [1]. Among marine botanical organisms, macroalgae have been deeply investigated and exploited; by contrast, the phytopharmaceutical profile of seagrasses is still almost unknown [2]. *Halophila stipulacea* is a tropical seagrass that colonized the Mediterranean Sea following the opening of the Suez Canal in 1869. This invasive species was reported for the first time on the Tunisian coast at Sfax (south-east Tunisia) [3]. Afterward, it was observed at Monastir (east Tunisia) by Sghaier et al. [4]. *Halophila stipulacea* can form seagrass assemblies up to 35 to 40 m deep in the Mediterranean, but is more often found in shallower habitats (2 to 10 m), in areas of low hydrodynamic turbulences, or near the ports. This species is mainly consumed by invertebrates, teleosts (fish), and locally by the turtle *Chelonia mydas*. Its establishment in the Mediterranean has allowed the introduction of associated exotic species such as the herbivorous aplysia *Syphonota geographica* [5]. Few scientific data on the bioactivities and therapeutic properties of secondary metabolites of *Halophila stipulacea* are available. Antioxidant and antibacterial activities have been observed in ethanolic leaves extracts [6]. Stems have never been evaluated, to the best of our knowledge, and no comparative studies are available between the bioactivity of leaves and stems.

In this context, this phytochemical study aimed to valorize *Halophila stipulacea* as a source of bioactive molecules for the treatment of several human diseases, as well as antifouling applications. Thus, organic extracts of *Halophila stipulacea* obtained by different polarities from leaves (L) and stems (S) (hexane [HL, HS], ethyl acetate [EL, ES], and methanol [ML, MS]) were evaluated for activity against human cancer cells and non-carcinogenic cells in monolayer culture and/or in multicellular spheroids. The potential bioactivity in metabolic diseases was assessed through the following assays: Glucose uptake in cells (diabetes), reduction of lipid content in fatty acid-overloaded liver cells (steatosis), and lipid-reducing activity in zebrafish larvae (obesity). Furthermore, more ecological applications of extracts were tested concerning the antifouling potential against microfouling (marine bacteria) and macrofouling (mussel larval settlement) species. Bioactivity data were complemented by the characterization of the metabolite profile of different extracts that were associated with observed bioactivities.

## 2. Results

### 2.1. Cytotoxic Activity

The cytotoxic activity of different extracts of leaves and stems of *Halophila stipulacea* was assessed against neuroblastoma SHSY5Y, colon adenocarcinoma HCT116, and osteosarcoma MG-63 cancer cell lines. The extracts EL, ES, HL, and HS reduced the viability in all three cell lines after 48 h exposure by approximately 50% at the concentration of 30 µg/mL (with the exception of HL and HS in MG-63 cells) (Figure 1). ML and MS extracts did not significantly inhibit cell proliferation. EC_50_ values were calculated in additional assays for the extracts HL, HS, EL, and ES in the cancer cell line with the highest activity and in the non-carcinogenic endothelial cell line hCMEC (Table 1). However, all active extracts showed a similar reduction of viability on the normal cell line hCMEC, and should be regarded as generally cytotoxic to mammalian cells. The extract HS was the only one to present a fourfold higher toxicity in cancer cells compared to normal cells after 24 h of exposure (selectivity index of 3.9).

Furthermore, the cytotoxicity of extracts was tested in a 3D cell culture of the HCT116 cell line, a relevant model of solid tumors. The extracts did not have any significant effect; however, the extract HL slightly increased the number of dead cells, as indicated by propidium iodide (PI) staining of the multicellular spheroid (Figure 2).

### 2.2. Lipid-Reducing Activity

The exposure of zebrafish larvae identified lipid-reducing activity of some extracts at 2 or 6 µg/mL (Figure 3). A significant decrease in Nile red staining was observed for EL and ML extracts with EC_50_ values of 2.2 µg/mL for EL and 1.2 µg/mL for ML after 48 h of exposure. Toxicity was evaluated on zebrafish larvae exposed to these extracts considering general toxicity (death after 24 or 48 h). No toxicity was observed for all the extracts at the tested concentrations. 

### 2.3. Anti-Steatosis and Anti-Diabetes Activities

The results of the anti-steatosis and anti-diabetic activities are shown in Appendix A. No significant effect on the reduction of lipids in the fatty acid-overloaded liver cells and on glucose uptake was observed for all the extracts. 

### 2.4. Anti-Fouling Activity

Regarding anti-fouling activity, the extracts HS and ML showed a significant inhibitory effect (*p* < 0.05) against the settlement of *M. galloprovincialis* plantigrade larvae (Figure 4) with EC_50_ values of 11.3 and 17.5 µg/mL for HS and ML, respectively. Regarding toxicity, no extracts caused mortality to the plantigrade larvae of the biofouling species *M. galloprovincialis* at the tested concentrations. The extracts did not have any significant effect against the growth of several species of marine fouling bacteria (Appendix A). 

### 2.5. Metabolite Profiling

Identification of the bioactive metabolites in the extracts was performed using HR-LC-MS-guided molecular networking analysis. The analysis led to the characterization of several known and unknown compounds by GNPS (The Global Natural Product Social Molecular Networking) analysis. The 110 identified metabolites (65 in positive mode, 45 in negative mode) from the extracts belong to different natural product categories, including primary metabolites (vitamins, fatty acids, sugars, amino acids, and peptides), secondary metabolites (polyphenols, chlorophylls, terpenoids, and alkaloids), and some synthetic compounds (Appendix A, and Figure 5). The detection of synthetic compounds can be explained by bioaccumulation and the described potential of this seagrass for phytoremediation [7]. Identified compounds with known functions related to the studied bioactivities are shown in Table 2. The quantification of selected mass peaks can be found in Appendix A. This selection was based on compounds being present in active and non-active extracts, or dereplicated compounds with a matching bioactivity. The aim of the quantification was to understand if a higher abundance was present in the active extracts. In extracts with anti-obesity activity (EL, ML), a unique mass peak was found at m/z 380.337, while in extracts with anti-biofouling activity (ML, HS), unique mass peaks were detected at m/z 900.569, 322.215, 250.144, 815.499, 584.806, 371.104, and 387.286. 

## 3. Discussion

Seagrasses share adaptive metabolic features with both terrestrial plants and marine algae, resulting in a phytocomplex possibly endowed with interesting biological properties [8]. This is the first study concerning the effect of *Halophila stipulacea* extracts on human diseases as cancer and metabolic diseases (obesity, diabetes, and steatosis), on its biofouling capacity, and one of the very few scientific reports about the biological properties of the *H. stipulacea* phytocomplex. We tested bioactivities of several extracts with different polarities (hexane, ethyl acetate, and methanol) obtained from leaves and stems of *H. stipulacea* (HL, EL, ML, HS, ES, and MS). The analysis revealed some potential of the seagrass for the discovery of lipid-reducing and anti-biofouling compounds. The metabolite profiling of these extracts allowed the identification of compounds in bioactive extracts whose reported functions relate to the studied bioactivities. This analysis also allowed the identification of unique mass peaks in bioactive extracts that do not have an identification in databases (potential unknown compounds). 

A review about natural products with anti-cancer activities revealed that polyphenols were the most described compounds with such properties [9]. The majority of the polyphenols detected in some extracts (ES and MS) were flavonoids such as luteolin, which was reported as an anticancer agent. Luteolin significantly decreased the cell viability of MG-63 cells: 35% at 2.5 µg/mL and 72% at 12.5 µg/mL [10]. In concordance to our data, a stronger mass peak of luteolin was present in ES (100%) compared to MS (67%), which might have contributed to the stronger reduction of viability in ES. Treatment of neuroblastoma SH-SY5Y cells with apigenin at 50 µM (13.5 µg/mL) led to the induction of apoptosis and reduced 30% of the cell viability [11]. In accordance, apiginin was identified in all extracts, but the highest cytotoxicity was noted in EL for SH-SY5Y cells, which clearly had the strongest peak intensity of apigenin (EL 100%, ES 17%, HL 8%, HS 4%, ML 16%, MS 31%). Previous studies have shown the detection of these flavonoids such as luteolin and apigenin in the *Hydrocharitaceae* family [12,13]. The polyphenol matairesinol was exclusively detected in HL and HS extracts, which was described to exhibit significant antiproliferative activity against CCRF-CEM cells with an IC_50_ of 4.27 µM (1.5 µg/mL) [14]. Lyngbyabellin A was detected in HL and HS, which was reported as a strong cytotoxic agent against the HCT116 colon cancer cell line with an IC_50_ of 40 nM [15]. Consistent with the presence of the known cytotoxic molecule, viability reduction of HL and HS was stronger on HCT116 cells. Previously, this compound has been isolated from marine *Cyanobacterium* species [16]. Here, for the first time, we detected lyngbyabellin A in *Halophila* species, but it remained unknown whether the compound was accumulated or produced by the seagrass. In general, observed cytotoxicity of the seagrass *Halophila stipulacea* was in concordance to other studies that demonstrated the cytotoxicity of cancer cells in response to extracts of seagrasses. Gavagnin et al. [17] reported that syphonoside from *H. stipulacea* inhibited apoptosis in some of the cell lines investigated. Moreover, crude extracts of the seagrass *Thalassodendron ciliatum* exhibited cytotoxic activity against HCT116 cells with an IC_50_ of 4.2 µg/mL [18]. However, as extracts of the seagrass *Halophila stipulacea* had a similar activity on a non-carcinogenic cell line and did not show activity on spheroids from the 3D cell culture, they are not regarded as interesting material for future works on anticancer activities.

Our work evidenced the lipid-reducing activity of some extracts (EL, ML) with EC_50_ values of 1.2–2.2 µg/mL using the zebrafish Nile red fat metabolism assay as a whole small animal model *in vivo*. This assay was used previously to show the lipid-lowering effects of extracts of raw materials or marine organisms (wine lee, actinobacteria, cyanobacteria, sponges), or isolated compounds (polyphenols, chlorophyll derivatives) [19,20,21,22,23]. The original paper from Jones et al. [24] demonstrated that zebrafish responded to lipid modulator drugs in the same way as humans. Additionally, the zebrafish model was compared to the cellular differentiation model of 3T3L1 preadipocytes and obese mice models, and all three assays demonstrated the same effect of chrysophanic acid, an anthraquinone from rhubarb [25]. A similar comparison was done for lipid accumulation after quercetin exposure in 3T3-L1 cells, zebrafish, and mice [26]. In summary, data indicate that the Nile red fat metabolism assay in zebrafish larvae is an interesting preclinical screening model in live whole organisms. Castro et al. [27] reported that plants are the main source of natural compounds with anti-obesity properties, and the majority of these compounds belong to the phenols and polyphenols class. The screening of a library of polyphenol derivatives identified compounds with strong lipid reducing activity in the zebrafish Nile red fat metabolism assay [20]. The naphthoquinone shikonin (5,6-dihydroxyflavone-7-glucuronic acid), a compound isolated from the Japanese plant *Lithospermum erythrorhizon*, inhibited fat accumulation in 3T3-L1 adipocytes [28]. Interestingly, the flavonoid cirsimarin was detected in our study (in EL, HL, and ML): 5-hydroxy-6,7-dimethoxy-2-[4-[(2S,3R,4S,5S,6R)-3,4,5-trihydroxy-6-(hydroxymethyl)oxan-2-yl]oxyphenyl]chromen-4-one. This flavonoid, known as the active compound of the plant *Microtea debilis*, was shown to exert potent antilipogenic effects and to decrease adipose tissue deposition in mice at 25 mg/kg [29]. It could, therefore, be one of the responsible metabolites for the observed lipid-reducing activity in the zebrafish; indeed, ML (100%) and EL (33%) showed higher peak intensities than HL (14%) for this mass peak. Another flavonoid was detected in ML and MS: Spiraeoside, which is the 4′-O-glucoside of quercetin, and a higher peak intensity was observed in ML (ML 100%, MS 67%). A previous study showed that quercetin inhibits lipid accumulation and obesity-induced inflammation using 3T3-L1 and zebrafish models at 25 µM (7.5 µg/mL), and mice at 100 mg/kg [26]. Likewise, a study conducted by Zhao et al. [30] suggested that the combination of quercetin and resveratrol reduced high fat diet-induced obesity and inflammation in rats. One polyphenol (2,4-dihydroxyheptadec-16-ynyl acetate) was exclusively present in the EL extract with lipid-reducing activity. Interestingly, this compound was shown to be an inhibitor of the acetyl-CoA carboxylase with an IC_50_ of 5.1 µM (1.6 µg/mL) [31,32], which is a key enzyme in fatty acid metabolism. Inhibition of this enzyme is regarded as an effective approach for treating the metabolic syndrome [32]. Furthermore, N-acetyl-L-tyrosine was exclusively present in the EL and ML extracts, which both had lipid-reducing activities. Derivatives of N-acetyl-L-tyrosine were shown to have binding affinity to the peroxisome proliferator-activated receptor alpha (PPARα) at 50 µM (9.1 µg/mL), which is a regulator of lipid homeostasis, and PPARα agonists are described to decrease hyperlipidemia [33]. Both compounds, unique to bioactive fractions EL and ML, may be responsible for at least a part of the observed lipid-reduction activity. Besides the mass peaks with a putative identification by GNPS-based database matches, a non-identified mass peak was observed exclusively in the EL and ML extracts (380.337 m/z), and this unknown compound may have contributed to the observed lipid reducing activities. Future works should focus on isolation of this mass peak in order to (i) test its bioactivity as a purified compound, and (ii) to elucidate its structures if bioactivity is observed. 

Our work also evidenced the antifouling activity of some extracts (HS and ML) against the efficient settlement of mussel plantigrade larvae. Iyapparaj et al. [34] reported that seagrasses are the submerged and sessile marine angiosperms that resist the attachment of epibionts. The same study also characterized the antifouling bioactive compounds of the extracts of the seagrasses *Syringodium isoetifolium* and *Cymodocea serrulata*, and identified a mixture of fatty acids as the active fraction of both species. Interestingly, 13-decosenamide is a fatty acid detected in EL, ML, HS, and MS, and, concordant to our bioactivity results, was most abundant in HS (HS 100%, MS 77%, ML 40%, EL 29%). This fatty acid (synonym: Erucamide) was reported as an antifouling agent against the settlement and adhesion of some marine organisms such as diatoms, microbial films, and abalones at a 4% concentration (118 mM; 39.8 mg/mL) [35]. Such a high concentration compared to the extract concentration used in this study (30 µg/mL) does not provide any evidence that 13-decosenamide is involved in the observed biofouling activity. Marine-sulfated secondary metabolites from seaweeds, namely, sulfated polyphenols, including cinnamic acids, are described as antifouling compounds compatible with the environment [36,37]. Zosteric acid, a metabolite isolated from the seagrass *Zostera marina* (a sulfated cinnamic acid), is recognized as a fully biodegradable and nontoxic natural antifouling agent [38]. Our study also showed the detection of these classes of compounds, such as 3-hydroxy-4-methoxycinnamic acid and alpha-cyano-4-hydroxycinnamic acid. However, the quantification of mass peak intensities along the analyzed extracts did not reveal a higher presence in any of the active extracts, which suggests that those compounds did not contribute to the observed antifouling activity. Furthermore, seven non-identified mass peaks were exclusively observed both in the HS and ML extracts (m/z 900.569; 322.215; 250.144; 815.499; 584.806; 371.104; 387.286). As the present compounds with known anti-biofouling activity seemed not to be involved, it is highly likely that these unknown compounds may have contributed to the observed antifouling activity.

## 4. Materials and Methods 

### 4.1. Plant Material

The seagrass *Halophila stipulacea* (Forsskål) Ascherson was collected from Marina Cap Monastir (Southern Mediterranean Sea) and immediately brought to the laboratory in plastic bags containing seawater to prevent dehydration of the plants. The seagrass was identified by Dr. Rym Zakhama Sraieb (High Institute of Biotechnology of Sidi Thabet) [4,39]. The collected specimens were washed thoroughly with tap water to remove all the extraneous matter such as epiphyte, sand particles, pebbles, and shells. The leaves and stems were separated from the roots. The samples were shade-dried at room temperature for five days until a constant weight was obtained and was then grounded in an electric mixer. The powdered samples were kept in air-tight containers and stored at −20 °C until further use.

### 4.2. Preparation of Seagrass Extracts 

Crude extracts of leaves (L) and stems (S) of *Halophila stipulacea* were prepared through sequential extraction using solvents of increasing polarities: Hexane (HL, HS), ethyl acetate (EL, ES), and methanol (ML, MS). Each sample was filtered with thin layered cotton. The filtrate was evaporated by a rotary evaporator, and the obtained extracts were stored at 4 °C until further analysis. Extracts were dissolved in DMSO at the stock concentrations of 6 and 2 mg/mL and used in the different bioassays. Final concentrations of extract in the assays were dependent on the chosen dilution in each assay, as well as the percentage of solvent control. 

### 4.3. Cell Culture

The human cell lines, SH-SY5Y (neuroblastoma) and HCT-116 (colon adenocarcinoma), were obtained from Sigma-Aldrich (St. Louis, Missouri, USA). MG-63 (osteosarcoma) and HepG2 (liver carcinoma) human cell lines were obtained from the American Type Culture Collection (ATCC) (Manassas, Virginia, EUA). Human endothelial hCMEC/D3 (human blood–brain barrier) cells were kindly donated by Dr. P. O. Courad (INSERM, France). All cell lines with the exception of SH-SY5Y were grown in Dulbecco’s modified Eagle medium (DMEM) from Gibco (Thermo Fisher Scientific, Waltham, MA, USA) supplemented with 10% fetal bovine serum (Biochrom, Berlin, Germany) and 1% penicillin/streptomycin (Biochrom) at 100 IU/mL and 10 mg/mL, respectively, and 0.1% amphotericin (GE Healthcare, Little Chafont, United Kingdom). The cell line SH-SY5Y was grown in a 1:1 mix of Eagle´s minimum essential medium (MEM) with the F12 Nutrient Mixture (Ham´s) medium, both from Life Technologies (Thermo Fisher Scientific), supplemented as described above, plus 1% of non-essential amino acids. Cells were grown in an incubator at 37 °C and 5% CO_2_.

### 4.4. Cytotoxicity on Cancer Cell Lines 

The cytotoxicity analysis of different extracts of leaves and stems of *Halophila stipulacea* was performed by using the MTT assay (3-(4.5-dimethylthiazole-2-yl)-2.5-diphenyltetrazolium bromide), which evaluates the mitochondrial dehydrogenase activity. Cells were seeded in 96-well plates at 1 × 10^4^ cells/cm^2^. After 24 h of incubation, the cells were treated with two different concentrations of plant extracts (final concentration of 15 and 30 µg/mL of extracts in cell medium; 1:200 dilution of stock concentration). After 24 or 48 h, cells were incubated for 4 h at 37 °C with 1 mg/mL MTT. The purple-colored formazan salts were dissolved in DMSO, and the absorbance was read at 550 nm in a multi-detection microplate reader (Biotek, Synergy HT). Three replicates were used per assay, and at least two independent assays were performed. Cytotoxicity was expressed as a percentage of cell viability considering 100% viability in the solvent control (0.5% DMSO). EC_50_ values were determined by dose–response curves in further assays by using a dilution series from 30 to 0.234 µg/mL in 8 dilution steps.

### 4.5. Cytotoxicity in 3D Cell Culture of HCT-116 Cell Line

For spheroid formation, 1 × 10^4^ HCT-116 cells were seeded per well in McCoy’s 5A medium, supplemented with 10% fetal bovine serum, 0.1% amphotericin, and 1% penicillin/streptomycin, in an ultra-low-attachment 96-well plate (Corning). The plates were incubated at 37 °C and 5% CO_2_. After 6 days of culture, the medium was replaced with fresh medium with different plant extracts (30 µg/mL) at the same incubation conditions for 48 h. Vehicle solvent control (DMSO) and positive control (Staurosporine, final concentration 500 nM) were included in the assay. Spheroids were stained with Hoechst 33342 (final concentration 5 µg/mL) for staining nuclei, with propidium iodide (final concentration 5 µg/mL) for the staining of dead cells, and with Calcein AM (final concentration 3 µM) for cellular esterase activity indicating viable cells [40]. For imaging, spheroids were analyzed with a fluorescence microscope (Leica DM6000B, Wetzlar, Germany). Fluorescence intensity was quantified in CellProfiler [41].

### 4.6. Zebrafish Nile Red Fat Metabolism Assay

Lipid-reducing activity of the different extracts was analyzed with the zebrafish Nile red fat metabolism assay, as described in [20]. Zebrafish adults and larvae were maintained under standard conditions at 28 °C. In brief, zebrafish embryos were raised from 1 DPF (days post-fertilization) in egg water (60 µg/mL marine sea salt dissolved in distilled H_2_O) with 200 µM PTU (1-phenyl-2-thiourea) to inhibit pigmentation. From 3 to 5 DPF, zebrafish larvae were exposed to extracts at a final concentration of 6 and 2 µg/mL in egg water (1:1000 dilution of stock concentrations) with the daily renewal of water and extracts in a 24-well plate with a density of 10–12 larvae/well (n = 10–12). A solvent control (0.1% DMSO) and positive control (REV, resveratrol, final concentration 50 µM) were included in the assay. Lipids were stained with Nile red overnight at the final concentration of 10 ng/mL. For imaging, the larvae were anaesthetized with tricaine (MS-222, 0.03%) for 5 min and analyzed in a fluorescence microscope (Leica DM6000B, Wetzlar, Germany). Fluorescence intensity was quantified in individual zebrafish larvae by ImageJ (http://rsb.info.nih.gov/ij/index.html). EC_50_ values were determined by dose–response curves in further assays by using a dilution series from 12 to 0.187 µg/mL in 7 dilution steps.

### 4.7. Glucose Uptake Assay 

The glucose uptake assay of different extracts of leaves and stems of *Halophila stipulacea* was performed using 2-NBDG in HepG2 cells (followed by MTT assay for analysis of viability) as described in [42]. Cells were seeded in 96-well plates at 1 × 10^6^ cells/well. After 24 h of incubation at 37 °C, the medium was replaced by Hank’s Balanced Salt Solution (HBSS). After incubation overnight at 37 °C, cells were treated with two different concentrations of plant extracts (final concentrations of 30, 10 µg/mL in cell medium; 1:200 dilution of stock concentration). A solvent control (0.5% DMSO) and positive control (Emodin, final concentration 10 µM) were included in the assay. Cells were stained with 2-NBDG (100 µL) in HBSS. After 1 h of incubation at 37 °C, the medium was changed twice to ice-cold HBSS, and the fluorescence was read at 465/540 nm (excitation/emission). Following this, cells were incubated for 2 h at 37 °C with 1 mg/mL MTT. The medium was replaced with DMSO and the absorbance was read at 570 nm.

### 4.8. Anti-Steatosis Assay

HepG2 cells were seeded in 96-well plates at 4 × 10^3^ cells/well and adhered overnight. The medium was changed to DMEM supplemented with 62 µM sodium oleate (Sigma-Aldrich, St. Louis, MO, USA) to induce the presence of lipid droplets [42] and two different concentrations of plant extracts (final concentration of 30, 10 µg/mL; dilution 1:200 of stock concentration). After 6 h, cells were stained with 75 ng/mL Nile red (Sigma-Aldrich) and 10 µg/mL Hoechst 33342 (Sigma Aldrich) in HBSS. After incubating at 37 °C for 10 min and in the absence of light, cells were washed twice with HBSS. Fluorescence was read in a Synergy HT multi-detection microplate reader (Biotek) at 485/572 nm excitation/emission for Nile red and 360/460 nm for Hoechst. Following this, cells were fixed with ice-cold trichloroacetic acid (TCA) for 1 h at 4 °C in the dark and then washed four times with ddH_2_O to remove all the TCA. After being air-dried, the cells were stained with sulforhodamine B (SRB) in 1% acetic acid for 15 min and rinsed quickly with 1% acetic acid five times to remove unbound SRB. Plates were air-dried and bound dyes were solubilized with 150 µL Tris–HCl (10 mmol/L, pH 10.5) before reading the absorbance at 492 nm with the reference at 650 nm in a multi-detection microplate reader (Biotek, Synergy HT).

### 4.9. Bioassay with Marine Fouling Bacteria

Five strains of marine fouling bacteria were purchased from the Spanish Type Culture Collection (CECT): *Cobetia marina* CECT 4278, *Vibrio harveyi* CECT 525, *Roseobacter litoralis* CECT 5395, *Halomonas aquamarina* CECT 5000, and *Pseudoalteromonas atlantica* CECT 570. The bacteria were inoculated in Marine Broth (Difco) medium at an initial density of 0.1 (OD600). Bacterial growth inhibition in the presence of the compounds was measured by the reading of optical density at OD600. The incubation time was 24 h and the incubation temperature was 26 °C. In addition, 96-well flat-bottom microtiter plates were employed (Orange Scientific), and each extract concentration, 3 and 30 μg/mL, was replicated four times. 

### 4.10. Mussel Larvae Anti-Biofouling Bioassays

Mussel (*Mytilus galloprovincialis*) juvenile (0.5 cm in diameter) aggregates were sampled in intertidal pools during low neap tides at Memória Beach, North Portugal (41^°^13’59” N; 8^°^43’28” W) and transported to the laboratory. Mussel plantigrade larvae (0.5–2 mm) were screened among the mussel aggregates in a binocular magnifier (Olympus SZX2-ILLT, Tokyo, Japan), isolated with filtered seawater and gently washed to remove adhered organic particles immediately before the bioassays. The selected *M. galloprovincialis* plantigrade larvae were further screened for the typical foot exploring behavior before bioassays. The anti-settlement bioassays were performed using 5 larvae per replicate in 24-well microplates with 4 replicates per condition during 15 h, at 18 ± 1 °C, and in the dark. Test solutions were prepared in filtered seawater (final concentration 30 µg/mL; 1:200 dilution of stock concentration). A negative control (0.5% DMSO) and a positive control (5 µM CuSO4) were included in all bioassays. After exposure, the anti-settlement activity was determined by the presence/absence of efficiently attached byssal threads produced by each individual larva for all the conditions tested, and the effect concentration 50% (EC_50_) that inhibited 50% of larval settlement was determined for selected bioactive extracts by dose–response curves in further assays by using a dilution series from 60 to 3.75 µg/mL in 5 dilution steps.

### 4.11. UPLC-HRMS/MS Analysis

For LC-MS analysis, the extracts were dissolved in UPLC/MS-grade methanol at a concentration of 1 mg/mL. Twenty microliters of each sample was separated on a Accela HPLC fitted with a Gemini C18 column (5 μm, 110 A, 4.6 mm ID × 150 mm, Phenomenex), coupled to an Accela PDA detector, Accela autosampler, and Accela 600 pump and to an LTQ Orbitrap TM XL hybrid mass spectrometer (Thermo Fischer Scientific, Bremen, Germany) controlled by an LTQ Tune Plus 2.5.5 and Xcalibur 2.1.0. (Thermo Scientific). The separation was carried out using a gradient of 100% solvent A (water with 0.1% *v/v* HCOOH) to reach 100% Solvent B (acetonitrile with 0.1% *v/v* HCOOH) during 25 min at a flow rate of 0.8 mL/min. Analysis was operated in positive and negative ion mode. The following parameters were used for data acquisition: The capillary voltage of the electrospray ionization source (ESI) was set to 3.1 kV for the positive mode and to 3.0 kV for the negative mode, and the capillary temperature was 300 °C. The sheath gas and auxiliary gas flow rate were at 40 and 10 (arbitrary units as provided by the software settings). The capillary voltage was 34 V for the positive mode and −35 V for the negative mode, and the tube lens voltage was 100 V for the positive mode and −150 V for the negative mode. Tandem mass spectrometry scans were obtained for selected ions with a mass range of 100–2000 Da. In order to evaluate the relative abundance of selected mass peaks, Xcalibur 2.1.0 software was used to quantify the mass peaks on the extracted ion chromatograms (XIC) for specific mass values and retention times.

### 4.12. Molecular Networking

The MS/MS data were converted from files (.raw) to .mzXML files using MSConvert software (http://www.proteowizard.sourceforge.net), and then submitted to the Global Natural Product Social Molecular Networking (GNPS) platform (http://gnps.ucsd.edu) [43]. The data were filtered by removing all MS/MS peaks within +/− 17 Da of the precursor m/z. The data were then clustered with MS-Cluster with a parent mass tolerance of 2 Da and a MS/MS fragment ion tolerance of 0.5 Da to create consensus spectra. Further, consensus spectra that contained less than 2 spectra were discarded. A network was then created where the edges were filtered to have a cosine score above 0.7 and more than 6 matched peaks. Further edges between two nodes were kept in the network only if each of the nodes appeared in each other’s respective top 10 most similar nodes. The spectra in the network were then searched against GNPS spectral libraries. The library spectra were filtered in the same manner as the input data. The data were then imported into Cytoscape v3.7.1 and visualized as a network of nodes and edges. 

### 4.13. Statistics

The data from bioactivities were represented as box-and-whisker plots with values in 5–95 percentiles. The Gaussian distribution was tested by a Kolmogorov–Smirnov normality test (*p* < 0.05), and the homogeneity of variance was determined by Bartlett’s test. If statistical assumptions were met, one-way analysis of variance (ANOVA) followed by Tukey’s test (parametric distribution) was performed; otherwise, Kruskal–Wallis followed by Dunn’s post-hoc test (non-parametric distribution) was used to compare the solvent control group (DMSO) and the extracts. Statistically significant differences were considered with *p*-values < 0.05. The data from dose–response curves were used to determine EC_50_ values for bioactivity level. Mean intensity fluorescence data were normalized to the mean values of the solvent control (100%) and to mean values of the positive control (0%), and concentrations of the extracts were log-transformed. A non-linear regression was applied with a variable slope and least-squares fitting to obtain the dose–response curves.

## 5. Conclusions

In our study, the extracts of *Halophila stipulacea* exhibited cytotoxic, lipid-reducing, and antifouling activities. In particular, the lipid-reducing activities in the low micromolar range were promising, and not associated with general toxicity or malformations in the zebrafish model. The metabolite profiling by mass spectrometry and analysis by GNPS identified various compounds with known activities both on cancer (polyphenols: Luteolin, apigenin, matairesinol), on metabolic diseases (polyphenols: Cirsimarin, spiraeoside, 2,4-dihydroxyheptadec-16-ynyl acetate; amino acids: N-acetyl-L-tyrosine), and on antifouling (fatty acids: 13-decosenamide; cinnamic acids: 3-hydroxy-4-methoxycinnamic acid, alpha-cyano-4-hydroxycinnamic). Additionally, non-identified mass peaks were likely to contribute to the observed bioactivities. For the first time, we demonstrated the potential of this invasive species as a source of bioactive metabolites and characterized the metabolites in terms of exploiting the abundant and untapped biomass represented by this plant in Tunisia. 

## Figures and Tables

**Figure 1 marinedrugs-18-00088-f001:**
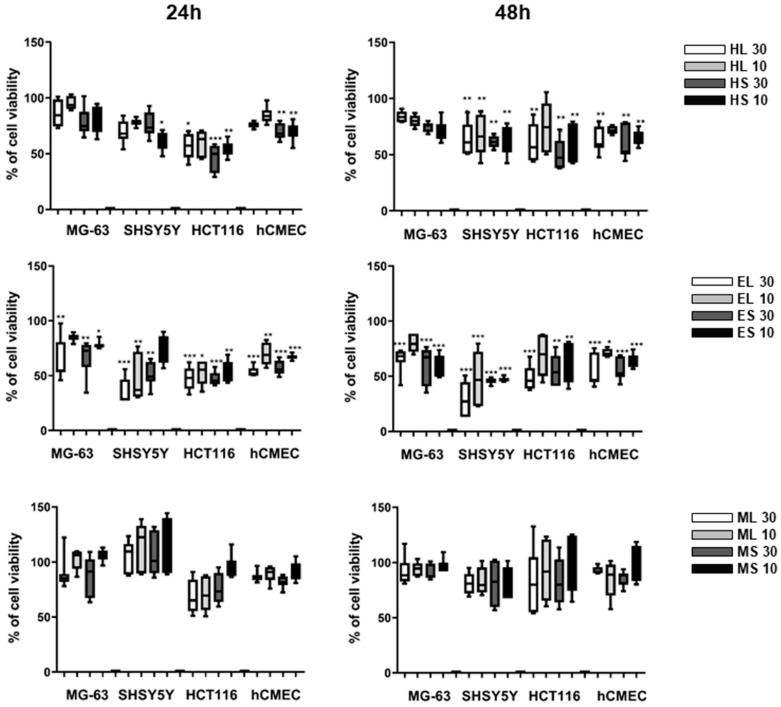
Cytotoxicity of different extracts at 10 or 30 μg/mL on human cancer cells (MG63, SHSY5Y, HCT116) and normal cells (hCMEC) after 24 and 48 h of exposure. The data were derived from two independent assays, each in triplicates, and represented in percentages relative to the solvent control. The data are represented as box-and-whisker plots (5–95 percentile) and statistical differences vs. solvent control are indicated by asterisks, * *p* < 0.05; ** *p* < 0.01; *** *p* < 0.001.

**Figure 2 marinedrugs-18-00088-f002:**
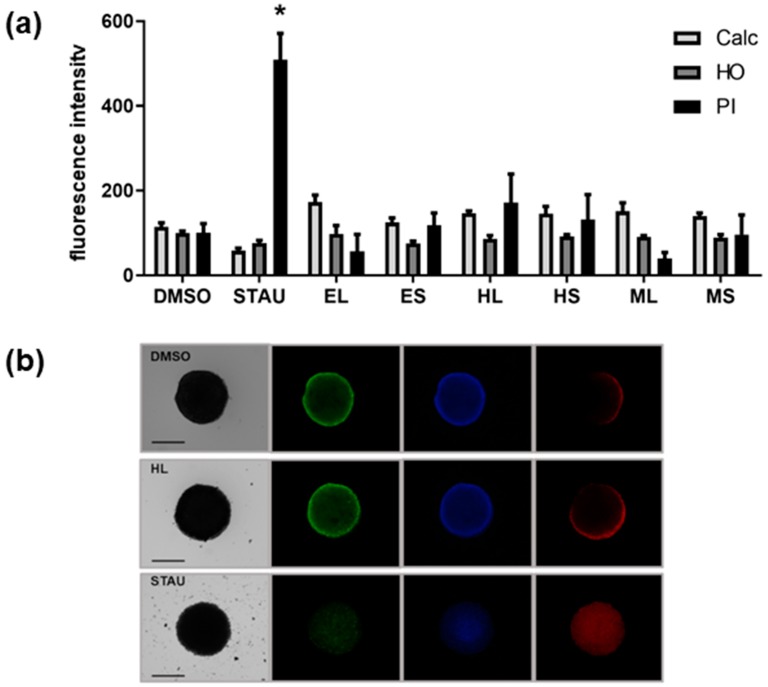
(**a**) Quantitative analysis of cytotoxicity in 3D cell culture of HCT-116 cell line of different extracts (30 µg/mL). DMSO, dimethylsulfoxide, solvent control; STAU, staurosporine, positive control; extracts (EL, ES, HL, HS, ML, MS); Calc, calcein AM staining for viable cells; HO, Hoechst 33342 staining for nuclei; PI, propidium iodide staining for dead cells. Statistical differences vs. solvent control are indicated by asterisks, * *p* < 0.05. (**b**) Representative images of spheroids under brightfield (**left**) and fluorescence microscopy (**right**). The scale bar shows 500 µm. Green fluorescence corresponds to calcein AM staining, blue fluorescence to Hoechst 33342 staining, and red fluorescence to propidium iodide staining.

**Figure 3 marinedrugs-18-00088-f003:**
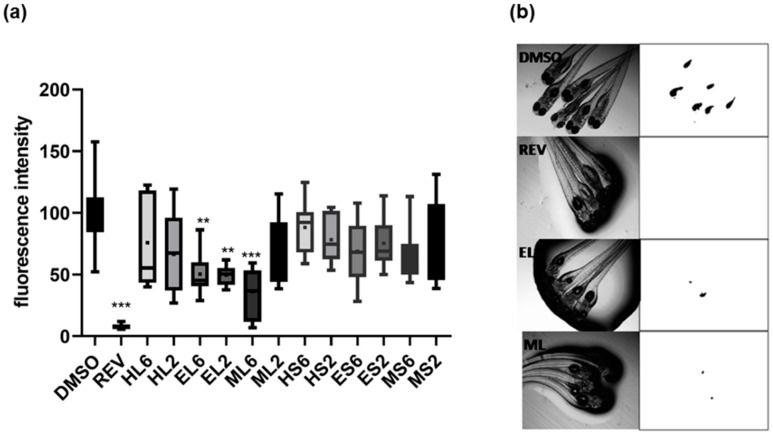
(**a**) Quantification of the lipid-reducing effects of different extracts at 2 or 6 µg/mL using the zebrafish Nile red fat metabolism assay. The data have been derived from 10–12 individuals per treatment group (n = 10–12) and are shown as box-and-whisker plots (5–95 percentiles). Statistical differences vs. solvent control are indicated by asterisks, ** *p* < 0.01; *** *p* < 0.001. (**b**) Representative images of zebrafish larvae under brightfield (**left**) and fluorescence microscopy (**right**). Red fluorescence is represented as inverted black and white images. DMSO, solvent control; EL, ML, extracts; REV, Resveratrol.

**Figure 4 marinedrugs-18-00088-f004:**
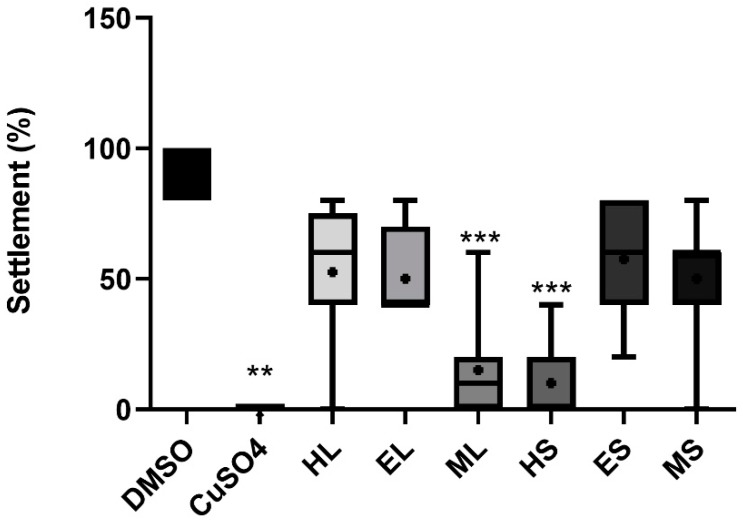
Anti-fouling activity of different extracts at 30 µg/mL toward plantigrade larvae of the mussel *Mytilus galloprovincialis*. The data have been derived from two independent assays each with four replicates per treatment group (n = 8). Each replicate consisted of five larvae. The data are shown as box-and-whisker plots (5–95 percentiles). Statistical differences vs. solvent control are indicated by asterisks, *** *p* < 0.001.

**Figure 5 marinedrugs-18-00088-f005:**
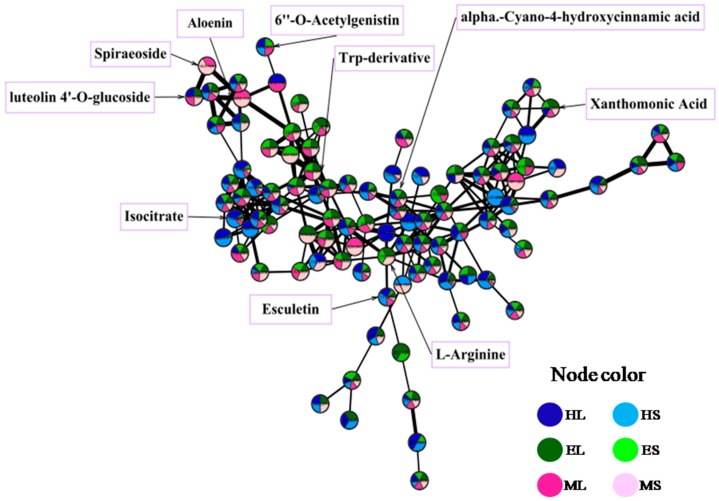
A representative cluster of molecular networking with GNPS, visualized using Cytoscape. Bigger edges represent a higher cosine score. The pie chart of each node shows the distribution of each mass peak along the six different seagrass extracts of this study. Some identified compounds are shown on the cluster.

**Table 1 marinedrugs-18-00088-t001:** EC_50_ values (µg/mL) of cytotoxic extracts in different cell lines, analyzed by the MTT assay.

Cell line	MG63		SHSY5Y		HCT116		hCMEC	
**Extracts**	**24 h**	**48 h**	**24 h**	**48 h**	**24 h**	**48 h**	**24 h**	**48 h**
**HL**					19.5 ± 5.8	29.1 ± 7.5	>30	>30
**EL**	>30	29.4 ± 6.3	10.6 ± 7.0	15.2 ± 1.9			11.3 ± 1.8	24.5 ± 15.6
**HS**					7.6 ± 5.4	25.4 ± 4.2	>30	>30
**ES**	>30	19.1 ± 9.0	23.4 ± 1.1	18.7 ± 3.1			9.2 ± 0.2	15.4 ± 1.1

**Table 2 marinedrugs-18-00088-t002:** Identified compounds with known functions that relate to the studied bioactivities. Identification was obtained from molecular networking with GNPS, based on LC-MS/MS data from different extracts (HL, EL, ML, HS, ES, MS). The color code highlights bioactivities (yellow: Cytotoxicity; blue: Obesity; green: Biofouling). In addition, 2,4-dihydroxyheptadec-16-ynyl acetate and N-acetyl-L-tyrosine were only present in active fractions with lipid-reducing activity (EL, ML).

Class	Compound Name	m/z	HL	EL	ML	HS	ES	MS
	Apigenin	271.228	x	x	x	x	x	x
	Luteolin	287.055					x	x
	Matairesinol	377.142	x			x		
**Polyphenols**	2,4-dihydroxyheptadec-16-ynyl acetate	325.184		x				
Spiraeoside	463.124			x			x
5-hydroxy-6,7-dimethoxy-2-[4-[(2S,3R,4S,5S,6R)-3,4,5-trihydroxy-6-(hydroxymethyl)oxan-2-yl]oxyphenyl]chromen-4-one	521.13	x	x	x			
alpha-Cyano-4-hydroxycinnamic acid	189.052	x	x	x	x	x	x
3-Hydroxy-4-methoxycinnamic acid	194.127	x	x	x	x	x	x
**Fatty Acids**	13-Docosenamide, (Z)	338.342		x	x	x		x
**Amino acids and peptides**	Lyngbyabellin A	713.473	x			x		
N-Acetyl-L-tyrosine	225.926		x	x

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
