# Peer review of "The Marine Seagrass Halophila stipulacea as a Source of Bioactive Metabolites against Obesity and Biofouling"

_marinedrugs, 2020, doi:10.3390/md18020088_

Round 1

Reviewer 1 Report

The authors have addressed my concerns from a previous submission, and have provided additional information which is quite useful. I have no further suggestions for the authors. 

Author Response

The authors have addressed my concerns from a previous submission, and have provided additional information which is quite useful. I have no further suggestions for the authors. 

Response: We thank the reviewer for its constructive review and help to improve the manuscript.

Reviewer 2 Report

Despite the attempts by the authors to imporve their manuscript, I still do not think that it is of standards to merit publication in Marine Drugs.

I still think that the paper can be better presented if split into two separate pieces of work.

Author Response

Despite the attempts by the authors to imporve their manuscript, I still do not think that it is of standards to merit publication in Marine Drugs.

I still think that the paper can be better presented if split into two separate pieces of work.

Response: We thank the reviewer for its constructive review and help to improve the manuscript. However, we still do not agree with the suggestion to split the manuscript into two pieces of work. In contrary, we believe that the evaluation of various bioactivities together with the characterization of metabolite profiles of seagrass extracts fits very well in the same work providing complementary information.

Reviewer 3 Report

The manuscript submitted for evaluation is an interesting and valuable study assessing the potential of active biomolecules of marine seagrass Halophila stipulacea. Studies in this area are particularly valuable because the therapeutic potential of biomolecules from marine organisms is still poorly understood and mostly undiscovered. Here are some comments:

In the second chapter of the work (results), the authors start by using abbreviations of extracts, while the reader has not yet been familiar with them. It would be valuable to enter short information about them in the introduction and create a list of all abbreviations. All abbreviations should be explained first where they appear. Separately in the abstract and main text. The HepG2 cells was not described in 4.3 Cell culture section of M&M Since the characteristics of cancer lines is at the end of the article, it is reasonable to introduce more precise descriptions eg., line no 73 "assessed against neuroblastoma SHSY5Y, colon adenocarcinoma HCT116, and osteosarcoma MG-63 cancer cell lines". Figure 1 and is not legible. Did the authors determine the lowest dose of extracts showing a cytotoxic effect on cancer cells but non-cancer cells? What was the basis for selecting of extracts doses? Were the lower doses not effective? It is valuable to determine the full spectrum for assessing a possible dose-dependence the lowest effective dose. The MTT test, read after 24 hours, actually examines the strong contribution of the proliferative component. Cytotoxicity tests using MTT are usually read much earlier, after just a few hours, to eliminate the overlapping effect of proliferation. In addition, the MTT test examines the effect on mitochondrial dehydrogenase activity, which itself can be modified by the administration of the test agent. Therefore, it is most appropriate to describe it in the context of activity or cytotoxicity but not viability. Authors should include this aspect in the description.

Author Response

The manuscript submitted for evaluation is an interesting and valuable study assessing the potential of active biomolecules of marine seagrass Halophila stipulacea. Studies in this area are particularly valuable because the therapeutic potential of biomolecules from marine organisms is still poorly understood and mostly undiscovered. Here are some comments:

In the second chapter of the work (results), the authors start by using abbreviations of extracts, while the reader has not yet been familiar with them. It would be valuable to enter short information about them in the introduction and create a list of all abbreviations. All abbreviations should be explained first where they appear. Separately in the abstract and main text.

Response: We added the requested information in the revised version.

The HepG2 cells was not described in 4.3 Cell culture section of M&M

Response: We added the requested information in the revised version.

Since the characteristics of cancer lines is at the end of the article, it is reasonable to introduce more precise descriptions eg., line no 73 "assessed against neuroblastoma SHSY5Y, colon adenocarcinoma HCT116, and osteosarcoma MG-63 cancer cell lines".

Response: We added the requested information in the revised version.

Figure 1 and is not legible.

Response: We improved the Figure 1 by increasing the size of the letters and a rearrangement of the graphics.

Did the authors determine the lowest dose of extracts showing a cytotoxic effect on cancer cells but non-cancer cells? What was the basis for selecting of extracts doses? Were the lower doses not effective? It is valuable to determine the full spectrum for assessing a possible dose-dependence the lowest effective dose.

Response: The selection of screening concentrations was done based on our past experience in cellular screening assays. We have selected two concentrations (30 ug/ml; 15 ug/ml) for evaluation of cytotoxic activity. Dose-response curves were done for most active extracts in cancer and non-cancer cells, which showed the expected dose-dependency and were used for the calculation of the presented IC50 values. 

The MTT test, read after 24 hours, actually examines the strong contribution of the proliferative component. Cytotoxicity tests using MTT are usually read much earlier, after just a few hours, to eliminate the overlapping effect of proliferation. In addition, the MTT test examines the effect on mitochondrial dehydrogenase activity, which itself can be modified by the administration of the test agent. Therefore, it is most appropriate to describe it in the context of activity or cytotoxicity but not viability. Authors should include this aspect in the description.

 Response: We agree with the reviewer´s opinion that MTT should be interpreted in the context of cytotoxicity. This was the interpretation that was used along the manuscript. We have added the information in the M&M section that the MTT test evaluates the mitochondrial dehydrogenase activity.

This manuscript is a resubmission of an earlier submission. The following is a list of the peer review reports and author responses from that submission.

Round 1

Reviewer 1 Report

This is an interesting manuscript showing the potential of seagrasses to provide bioactive compounds.

Is “lipid reducing activity in zebrafish larvae” a validated measure of obesity?

The Discussion must include literature studies that show these models used in this study are valid representations of the human diseases and predict new therapeutic interventions.

Lines 25 and 114: what are “fattened liver cells”?

Line 27 and 28; The cytotoxicity against cancer cell lines is correctly noted, but the key point is that this toxicity was only selective compared to normal cells with the HS extract.

The Abstract includes models of diabetes and liver steatosis, but no results are given; lines 113-115 report that there were no significant effects so this information should be in the Abstract.

Section 2.1: Please do not include data in both Table and text.

Of the range of compounds present in this seagrass, which ones, for example luteolin, have reported biological responses at similar concentrations?

A key parameter for bioactive compounds is the dose that is necessary to achieve the responses. What were the doses of the most likely bioactive compounds in the studies reported here? Are these doses similar to other biological activities of these compounds reported in the scientific literature, especially relating to cancer, diabetes and obesity?

Reviewer 2 Report

Reviewer Report on the manuscript titled: “The marine seagrass Halophila stipulacea as a source 2 of bioactive metabolites against obesity and biofouling” by Mabrouk SB et al. submitted to Marine Drugs

General comment

The manuscript by Mabrouk SB et al. is an interesting piece of work, however, it suffers from several methodological and presentational issues that make it unsuitable for publication in its present form.

I propose that the paper be split into two separate manuscripts, one, the first, describing the UPLC-HRMS/MS analysis and the Molecular networking analysis extensively and the other, the second one, describing the several bioassays the authors performed. Both manuscripts can be enriched with additional data that the authors have or should generate (see below).

As it stands, the present manuscript is very convoluted, confusing and stops short of identifying which compound in the extracts, they have isolated, does what. Because of this lack of identification, the reader gets very confused as to which compound in the extracts, is relevant for which assay/effect.

For example, if they speculate that compound X is probably responsible for the cytotoxicity, then why not try this compound X in its purified form to corroborate your claim and even provide quantitative data as to the % abundance of this compound X in the extracts.

In other words, what is missing from their methodological approach is a statement such as: “Y μg of Compound X are present in, for example, 30 μg of the extract and this amount is responsible for Z% cytotoxicity of the extract”. In this way, any reader will know exactly what the extract contains that make it cytotoxic. This is easily done since they quantified the compounds with Xcalibur (see line 365).

Major Comments

The authors use what appears to be arbitrary doses of extracts for the different assays. One, then, does not know if the doses used as really toxic for the cells or for the mussels. The fact that same doses of extracts were cytotoxic but did not inhibit glucose uptake is irrelevant here because the negative control (DMSO) was used in different concentrations for each assay. Following from above, one observes that different concentrations of the negative control (DMSO) were used in different assays that gave positive or negative results and this is deeply problematic. Why use different concentrations of DMSO as negative control? There is no rationale provided, in most of the assays, on why a given dose of extracts was used. Since there is, mostly, no data on dose-response of the extracts, it is very confusing to just state, for example, 30 μg/mL (mL of what?) of extracts. The data presented in Table 1 is very unclear where it came from. Did the authors determine the EC50 values from just two doses? If they did not, where is the dose-response assay data?

Minor comments

The authors do not really help their case by providing speculations and giving references in the Discussion section as to which compound in their extracts is responsible for what effect. Instead of just giving references, just corroborate your claims by using one of these compounds you describe in the Discussion section. Be very clear about the reporting of the negative and positive controls and the reasons why different concentrations of extracts were used. Also, the quality of some Figures is very poor, Figure 2a for example, where not explanation is given on the figure legend for the abbreviations. Figure 5 is very confusing and instead of simplifying things, it makes it worse. Cytoscape is too cool for your data, use another form of presentation.

Reviewer 3 Report

I applaud the efforts of authors, but I have major concerns.

The lipid reducing capacity does not match the anti fatty liver action and the lack of action on glucose metabolism, which are the mainstream of obesity.

Thus, the eventual anti obesity effect of these extracts is doubtful.